# Effect of Six Lactic Acid Bacteria Strains on Physicochemical Characteristics, Antioxidant Activities and Sensory Properties of Fermented Orange Juices

**DOI:** 10.3390/foods11131920

**Published:** 2022-06-28

**Authors:** Qi Quan, Wei Liu, Jiajing Guo, Meiling Ye, Juhua Zhang

**Affiliations:** 1Longping Branch Graduate School, Hunan University, Changsha 410125, China; qiquan@hnu.edu.cn; 2Hunan Agricultural Product Processing Institute, Hunan Academy of Agricultural Sciences, Changsha 410125, China; liuwei0505@126.com (W.L.); guojiajing1986@163.com (J.G.); 3Hunan Province Key Laboratory of Fruits & Vegetables Storage, Processing, Quality and Safety, Changsha 410125, China; 4Central Laboratory, Hunan Renzhi Testing Technology Co., Ltd., Changsha 410300, China; yeyefeiyang0806@126.com

**Keywords:** orange juice, lactic acid fermentation, physicochemical characteristics, antioxidant activities, volatile profiles, sensory properties

## Abstract

Six lactic acid bacteria strains were used to study the effects on physicochemical characteristics, antioxidant activities and sensory properties of fermented orange juices. All strains exhibited good growth in orange juice. Of these fermentations, some bioactive compositions (e.g., vitamin C, shikimic acid) and aroma-active compounds (e.g., linalool, 3-carene, ethyl 3-hydroxyhexanoate, etc.) significantly increased in *Lactiplantibacillus plantarum* and *Lactobacillus acidophilus* samples. DPPH free radical scavenging rates in *L. plantarum* and *Lacticaseibacillus paracasei* samples increased to 80.25% and 77.83%, respectively. Forty-three volatile profiles were identified, including 28 aroma-active compounds. 7 key factors significantly influencing sensory flavors of the juices were revealed, including D-limonene, linalool, ethyl butyrate, ethanol, *β*-caryophyllene, organic acids and SSC/TA ratio. The orange juice fermented by *L. paracasei*, with more optimization aroma-active compounds such as D-limonene, *β*-caryophyllene, terpinolene and *β*-myrcene, exhibited more desirable aroma flavors such as orange-like, green, woody and lilac incense, and gained the highest sensory score. Generally, *L. paracasei* fermentation presented better aroma flavors and overall acceptability, meanwhile enhancing antioxidant activities.

## 1. Introduction

Oranges are one of the most consumed fruits in the world owing to their unique orange aroma flavor and rich nutrients, e.g., carbohydrates, organic acids, amino acids, vitamins and minerals [1]. Meanwhile, an orange contains a variety of bioactive compositions such as vitamin C, flavonoids, limonoids and carotenoids which have been associated with their health benefits [2]. Although China is one of the important origins of oranges and it is mainly sold as fresh and processed into juice, the processing technology is relatively lagging and cannot meet the growing market demand. Moreover, the bitter taste after processing influences the sensory acceptability of orange juice to a certain extent. Furthermore, oranges suffering from mechanical injuries and even rottenness during the transportation or storage processes cause serious economic losses to fruit farmers and food companies. Therefore, it is essential to develop new orange processed foods to increase the storability, shelf life and added value.

Studies showed that lactic acid fermentation (LAF) could effectively improve the shelf life, nutrition and sensory quality of fruits and vegetables [3]. Lactic acid bacteria (LAB), the most commonly used strains for LAF, have been employed to produce a wide range of fermented juices [4,5]. For instance, *Lactiplantibacillus plantarum* often exists in fermented fruit and vegetable juices with various functions to enhance human immunity and promote nutrient absorption [6]. *Limosilactobacillus fermentum* could promote the production of volatile compounds in fruit and vegetable juice and improve the flavor quality [7]. *Levilactobacillus brevis* exhibited greater adaptability and fewer effects on the appearance of fermented bog bilberry juice [8]. Moreover, researchers found that different microorganisms had different impacts on the fermentation of watermelon juices and there was a linkage between sensory and volatile profiles [9]. However, it is well-known that the growth and fermentation of lactic acid bacteria are easily affected by various factors such as individual adaptability, plant matrices, and fermentation method, which further affect the quality and flavor of fermented products [10].

In the past, the majority of studies indicated that LAB was used to investigate the effects on the nutritional characteristics, volatile profiles and sensory properties of fruit and vegetable juices, so that improved the food properties. For instance, Al-Sahlany et al. found that onion inoculated with three kinds of LAB increased antioxidant performance, reduced total acidity, and had mutation-resistance and good taste and flavor [11]. Similarly, orange juice was fermented by LAB and some physicochemical characteristics and functional properties were also noted. For example, the addition of *Lactobacillus* spp. modified considerably the phenolic content of ‘Tarocco’ and ‘Washington navel’ juice, and the changes were contingent on both the quality of raw juice and the LAB employed [5]. In addition, *Citrus* juice fermented by *L. plantarum* and *L. pentosus* had increased lactic acid content and decreased *L*-malic acid as well as specific free amino acids [1]. Moreover, the orange-juice milk beverages fermented by *L. brevis* and *L. plantarum* were observed to have higher total polyphenols, total carotenoids and total antioxidant capacity [12]. However, some vital problems in orange juice fermentation, e.g., microbial adaptabilities, the relationship between sensory properties and LAB fermentation, etc., are currently studied in to a lesser extent. Therefore, we hypothesized that lactic acid fermentation could improve the functional and sensory properties of orange juice. For this purpose, six representative types of LAB, including *Lactiplantibacillus plantarum* (Lp), *Limosilactobacillus fermentum* (Lf), *Lactobacillus acidophilus* (La), *Lacticaseibacillus rhamnosus* (Lr), *Lacticaseibacillus paracasei* (Lc) and *Bifidobacterium longum* (Bl), were used to ferment orange juices and study the changes in terms of the physicochemical characteristics, antioxidant activities and sensory properties of fermented orange juices to clarify these questions in this study, which could provide a scientific basis for the deep processing of orange and other fruit and vegetable juices.

## 2. Materials and Methods

### 2.1. Lactic Acid Bacteria Strains

*L. rhamnosus*CICC20259 (Lr), *L. paracasei*CICC20251 (Lc) and *B. longum*CICC6188 (Bl) were obtained from the China Center of Industrial Culture Collection (Beijing, China). *L. plantarum* (Lp), *L. fermentum* (Lf) and *L. acidophilus* (La) were prepared by the Key Laboratory for Fruits and Vegetables storage Processing and Quality Safety in Hunan Province. According to accessible scientific evidence, probiotic strains typically provide fitness benefits by enhancing or restoring the gut microbiota when ingested [13,14]. They are permitted as healthy practical meal elements via China Health Inspector Food and Drug Safety Ministry in China. And the safety of strains must be evaluated when used as new food raw materials for mass production in factories or enterprises.

### 2.2. Reagents and Standards

De Man, Rogosa and Sharpe (MRS) medium, Modified MRS medium base were purchased from Guangdong Huankai Microbial SCI. & Tech. Co., Ltd. (Guangzhou. China). Reinforced medium for Clostridia (RCM), Reinforced Clostridium Agar were purchased from Qingdao Hope Bio-Technology Co., Ltd. (Qingdao, China). All reagents and solvents were analytical or HPLC grade. Sugar standards (glucose, fructose, sucrose, sucrose, lactose), organic acid standards (l-lactic acid, oxalic acid, acetic acid, citric acid, pyruvic acid, malic acid, shikimic acid, succinic acid, quininic acid, tartaric acid), vitamin C standard, flavone standards (naringin, hesperidin, didymin, neohesperidin, poncirin, sinensetin, rhoifolin, nobiletin) and limonin standard were obtained from Shanghai Aladdin Biochemical Technology Co., Ltd. (Shanghai, China) & Chengdu DeSiTe Biological Technology Co., Ltd. (Chengdu, China). Acetonitrile, phosphoric acid, methanol (Shanghai, China) were prepared for ultra-performance liquid chromatography (UPLC) analysis. For gas chromatography spectrometry (GC-MS), cyclohexanone diluted 40 times with methanol (Shanghai, China) was used as an internal standard. 

### 2.3. Sample Preparation

Natural ‘Langshan’ navel orange juice was purchased from a company (Xinning, China) and treated by high-pressure processing (HPP), preserved at –30 °C. Before juice was added, *Lactobacillus* strains were activated in MRS and *Bifidobacterium* strain in RCM broth to about 7.0 log CFU/mL, respectively. HPP ‘Langshan’ navel orange juice (11.24 °Brix; pH 4.23; with no viable cells) was filtered by filter cloth (100 mesh) and adjusted to pH 6.6 with food-grade KHCO_3_. Referring to the previously reported method and modified appropriately [8,15], the juice samples were separately inoculated with *L. plantarum* (Lp), *L. fermentum* (Lf), *L. acidophilus* (La), *L. rhamnosus* (Lr), *L. paracasei* (Lc), *B. longum* (Bl) at 9:1, 7:3, 5:5, 3:7, 1:9 gradient successive acclimation, and cultured at pH 6.6, 37 °C for 36 h. After that, the cells were harvested by centrifugation at 5000× *g* rpm for 10 min at 4 °C using Avanti J-26 XP centrifuge (Beijing, China). Then, they were washed twice with sterilized 0.9% NaCl solutions and inoculated in juice at 3%, cultured at pH 6.6, 37 °C for 12 h to obtain seed liquid. Finally, 1 mL seed liquid was added to 100 mL juice sample, and subsequently fermented at 37 °C pH 6.6 for 48 h. After fermentation, the samples were cooled in an ice bath and stored at –30 °C before analyses. Original orange juice served as the control sample.

### 2.4. Physicochemical Characteristics and Viable Cells Tests

#### 2.4.1. Viable Cell Counts

The number of viable lactic acid bacteria in samples was determined on MRS AGAR (Guangzhou, China) by standard tandem dilution method. Under anaerobic conditions for 48 h (37 °C), the specific operation was to dilute 1 mL of bacterial liquid or fermentation liquid to an appropriate gradient, and spread 100 μL on MRS AGAR (Guangzhou, China), respectively. Culture at 37 °C for 48 h, 30–300 CFU plates were selected for counting, and each group was repeated 3 times [16]. 

#### 2.4.2. pH, Total Acids (TA) and Soluble Solid Content (SSC)

The pH of all samples was determined with a PHS-3C pH meter (Shanghai, China). The soluble solid content (SSC) was determined using a hand-held refractometer (LYT-380) and expressed as °Brix [17]. The total acid (TA) content was determined by potentiometric titration and calculated based on the conversion coefficient of lactic acid. Each juice sample was titrated with standardized aqueous NaOH solution (0.1 M) and phenolphthalein solution was added as an indicator. In order to exclude the influence of fruit juice color on titration, pH value 8.20 ± 0.01 was set as the endpoint of titration, and each juice sample reached and maintained 4–5 s. The consumption volume of standard NaOH aqueous solution (0.1 M) was recorded for the determination of TA content [1,18].

#### 2.4.3. UPLC Determination of Soluble Sugars and Organic Acids

Soluble sugars and organic acids were detected by UPLC on a Shimadzu UPLC system (DGU-20A, Kyoto, Japan). Before analysis, the sample (5 mL) was diluted 10 times and filtered through a 0.45 μm hydrophilic filter membrane and 10 μL of the solution was injected into the UPLC system. The separation columns for sugars and organic acids were a Inertsil NH_2_ column (250 mm × 4.6 mm, pore size 5 μm) and a XY-C18 column (250 mm × 4.6 mm, pore size 5 μm), respectively. Acetonitrile: distilled (70:30) water was used as mobile phases with a flow rate of 1.0 mL/min for sugar analysis, 0.01 mol/L KH_2_PO_4_-H_3_PO_4_ (pH 2.7) and methanol were applied as mobile phases at a flow rate of 0.7 mL/min for organic acids analysis [19]. Sugars and organic acids concentrations (g/L) were calculated using standard curves.

#### 2.4.4. UPLC Determination of Vitamin C (Vc) and Flavones

The contents of Vc and flavones in juice samples were measured by UPLC (Shimadzu, Kyoto, Japan). For Vc content analysis, 10 mL of juice samples were mixed with 40 mL of oxalic acid (1%) and subsequently 20 μL of obtained solution were injected into a C18 column (250 mm × 4.6 mm, pore size 5 μm). KH_2_PO_4_ (0.025 mol/L) was used as a mobile phase with a flow rate of 1 mL/min [20]. For flavones contents analysis, 5 mL juice samples were extracted with methanol (80%) and dimethyl sulfoxide (1:1, *v*/*v*). After filtration on 0.22 μm organic membrane (Syringe filter), 10 μL of the solution was injected into the UPLC system. 0.1% formic acid (aqueous) and methanol were used as mobile phases with a flow rate of 0.7 mL/min. Vc (mg/100 g) and flavones (g/L) concentrations were calculated using standard curves [21].

#### 2.4.5. UPLC-MS-MS Determination of Limonin

The content of limonin in juice samples was measured by ultra-high performance liquid chromatography-tandem mass spectrometry (UPLC-MS-MS, Waters, Milford, MA, USA). Before analysis, the sample (5 mL) was extracted with acetonitrile (10 mL), vortexed for a while, ultrasonic for 30 min, and filtered through 0.22 μm organic membrane. The mobile phase consisted of acetonitrile-water (A:B) with a flow rate of 0.35 mL/min, and the programmed gradient elution was performed as follows: A 10%(0.5 min)–50%(2.5 min)–50%(3 min)–90%(4 min)–90%(4.5 min). The ionization mode was atmospheric pressure electrospray ion source positive ion mode (ESI+), collision cell voltage was 30 V, collision argon flow rate at 0.14 mL/min. Multiple response monitoring was used, and the parent ion *m*/*z* = 471.1, daughter ion *m*/*z* = 425.3, *m*/*z* = 161.1. Limonin concentration (g/L) was calculated using standard curves.

### 2.5. HS-SPME-GC-MS Determination of Volatile Compounds

All volatile compounds were analyzed by 7890A-5975C gas chromatography-mass spectrometry (GC-MS, Agilent, Santa Clara, CA, USA) with headspace solid-phase micro-extraction (HS-SPME, 50/30 μm, CNW Germany, Frankfurt, Germany). Briefly, 1.7 g NaCl was added to a 15 mL headspace extraction bottle, followed by the addition of 10 mL sample and 100 μL internal standard (methanol diluted 40 times cyclohexanone). The volatile compounds were separated on the DB-5MS capillary column (30 cm × 0.25 mm, 0.25 μm) using helium as the carrier gas at the flow rate of 1.2 mL/min. The programmed gradient elution was performed as follows: 40 °C for 0 min, then increased to 100 °C at 3 °C/min, to 200 °C at 5 °C/min, and finally to 260 °C at 10 °C/min (maintained for 0 min). Mass spectrometry used electron ionization at 70 eV, ion source and transfer line temperatures were 230 °C and 280 °C, respectively. Scanning mode was adopted for detection, and the detection range was between 50–550 *m*/*z* [16].

### 2.6. Antioxidant Activities Assay

#### 2.6.1. DPPH Radical Scavenging Activity

The 2,2-diphenyl-1-picrylhydrazylradical scavenging activity (DPPH RSA) of the fermented samples was evaluated using the method with some modifications [22]. A volume of 20 μL of different juice samples was mixed with 380 μL of a methanolized solution of DPPH radical and reacted in the dark for 20 min. The absorbance (A) was measured at 515 nm using a BioTek Synergy H1 microplate reader (Agilent, Santa Clara, CA, USA), with a mixture of 20 μL extract and 380 μL DPPH solution as blank control. DPPH free radical scavenging rates of the samples were calculated as follows:DPPH RSA (%) = (*A*_control_ − *A*_sample_)/*A*_control_ × 100

#### 2.6.2. ABTS Assay

A mild modification of the approach was used to assess the 2,2′-azino-bis(3-ethylbenzothiazoline-6-sulfonic acid) radical scavenging activity (ABTS RSA) of the samples [17]. In essence, 10 μL of samples were added to 190 μL of ABTS mixture solution. Thereafter, the mixture was incubated in the dark for 10 min and the absorbance (A) was measured at 734 nm using a microplate reader (BioTek Synergy H1, Agilent, Santa Clara, CA, USA). ABTS RSA rates of the samples were calculated as follows: ABTS RSA (%) = (*A*_control_ − *A*_sample_)/*A*_control_ × 100

#### 2.6.3. FRAP Test

The ferric-reducing antioxidant powers (FRAP) of the fermented samples were measured according to the previously reported method with slight modifications [17]. In short, 10 μL of juice samples were added to 190 μL of the reaction mixture solution. The mixture consisted of 0.3 mol/L acetic acid buffer salt, 10 mmol/L 2,4,6-Tri(2-pyridyl)-1,3,5-triazine and 20 mmol/L FeCl_3_ solution. The mixture was consequently incubated at 37 °C in the dark for 20 min and the absorbance (A) was measured at 593 nm using a microplate reader (BioTek Synergy H1, Agilent, Santa Clara, CA, USA). The reducing power was expressed as mmol Trolox/L using Trolox as a standard.

### 2.7. Sensory Evaluation

Sensory measurements of juice samples were assessed by a sensory descriptive analysis (DA) and evaluated by 20 teachers and students panel members with relevant experience and background knowledge [23]. Each panelist received seven samples and evaluated them on 9 characteristics in a sensory room. And the evaluating details included four parts: appearance (orange color, light color), aroma (fermented, green, orange aroma), taste (sweetness, sourness, bitterness) and overall acceptability, which were previously agreed as most important sensorial characteristics of juices [24]. The intensity and likeability of samples were rated on a scale of 1–10 (1–2 = very weak intensity, dislike slightly, 3–4 = weak intensity, like slightly, 5–6 = moderate intensity, like moderately, 7–8 = strong intensity, like very much, and 9–10 = very strong intensity, like extremely), by using a list of descriptors. The results of the sensory evaluation analysis for each odor note were calculated based on the average scores of 20 panel members and plotted in a sensory radar diagram.

### 2.8. Statistical Analysis

All assays were carried out in triplicates and the data were presented as mean ± standard deviation. The analysis of variance (ANOVA), correlation analysis and stepwise multiple regression analysis were performed using IBM SPSS 25.0. The correlation analysis was conducted with Pearson correlation coefficient method. The key factors significantly influencing sensory properties and aroma-active compounds of fermented orange juices were determined with stepwise multiple regression analysis and odor activity values (OAVs) of volatile compounds [25,26]. The clustered heat map, principal component analysis (PCA) and sensory radar diagram were plotted using Origin Pro version 2018 (Origin Lab, Northampton, MA, USA). *p* < 0.05 was considered statistically significant.

## 3. Results and Discussion

### 3.1. Detection and Analysis on Physicochemical Characteristics of Fermented Orange Juices and Viable Count

#### 3.1.1. Viable Count Detection

Cell viability, as an important factor in evaluating functional products [27], was focused in this study. Briefly, the six tested LAB strains were activated to about 7.0 log CFU/mL under the same suitable conditions before added into juices. All six LAB strains exhibited good growth in orange juices at pH 6.6, 37 ℃ for 48 h, with viable counts of 7.42–8.07 log CFU/mL (Table 1). The result showed that orange juice, as a lactic fermentation substrate, could be beneficial for the growth of LAB, whose concentration was always higher than the minimum to maintain a healthy life (7.0 log CFU/mL) [16].

#### 3.1.2. Variations in Soluble Solid Content (SSC) and Soluble Sugars

Sugars, also known as carbohydrates, are one of the important components in orange juice and an important carbon source for microbial growth [28]. As listed in Table 1 and Figure 1, after LAB fermentation, the contents of soluble solid (SSC), soluble sugars except of glucose in orange juices significantly decreased compared to control, in which the SSC in all samples significantly decreased to 8.05–8.85 °Brix (*p* < 0.05). The contents of sucrose in Lp sample, fructose and maltose in La sample extremely decreased to 12.31, 2.68 and 1.85 g/L, respectively. Meanwhile, the total content of soluble sugars in Lp sample was the lowest, decreasing by 40.59% compared to control after the fermentation. Except of Lr sample, the contents of glucose in the other five samples increased by 17.48–25.05% compared to the control. Therefore, different LAB strains had the common characteristics of utilizing sugars, but also presented their own unique characteristics of metabolizing sugars. A similar study showed that LAF produced a large amount of lactic acids in juice, provided a low pH environment, speeded up the hydrolysis rate of sucrose into glucose and fructose, which was faster than the rate of sugar consumption, thus resulting in the increase of glucose [28]. Meanwhile, Hashemi et al. also reported that the consumption of sugar by microorganisms was closely related to cell strain, fermentation substrate and other factors [22]. These results further confirmed the metabolism characteristics of the soluble sugars in the processing of the LAF.

#### 3.1.3. Variations in Total Acid (TA) and Organic Acids

Organic acids, as the most important components to balance the complex and diverse aromatic flavors in fruits, were closely related to the acidity and flavor characteristics of juice, and had some influences on the microbial community structure, fermentation rate and shelf life of orange juice [29]. As shown in Table 1, the contents of TA in all samples significantly increased by 10.14–78.26% over the control (*p* < 0.05), while the pH values were drastically decreased from 6.60 to 3.96–4.82, leading to the SSC/TA ratios of the fermented orange juices significantly reduced from 16.30 to 6.70–11.12 (*p* < 0.05). Because of the growth and metabolism of lactic acid bacteria, which consumed the original (such as citric acid) or produce organic acids (such as lactic acid) in the fermentation process, resulting in the increase of TA and the decrease of SSC/TA ratio [30].

To further explore the changes of organic acids before and after fermentation, 10 organic acids were detected (Figure 2). It was indicated that malic acid, citric acid, shikimic acid, lactic acid and acetic acid were identified and varied remarkably, but tartaric acid, quinic acid, oxalic acid, pyruvic acid and succinic acid were not detected. As shown in Figure 2, citric acid was the richest organic acid in original orange juice (OJ), accounting for 83.37%, but its concentrations in all fermented samples significantly reduced by 17.70–32.74% compared with the control (*p* < 0.05). Especially in Lp, La and Lr fermented samples, the contents of citric acid decreased to 2.28, 2.42, 2.43 g/L, respectively. The earlier studies found that during the lactic acid fermentation, citrate was metabolized with pyruvate as the central intermediate, utilizing two different pathways (e.g., pyruvate/formate lyase) leading to directly causing the production of acetate and ATP [31]. The acetic acid contents in 6 fermented samples ranged from 1.94 to 2.47 g/L, but it was not found in original juice. These results further proved that acetic acid was produced during the LAB fermentation. Lactic acid is an important indicator to evaluate the success of fermentation. In this study, all tested strains showed the ability to produce a large amount of lactic acid, in which lactic acid in Lr and Lc samples were higher (6.65 and 6.18 g/L, respectively) than that in other samples. But malic acid contents in all fermentations significantly declined (*p* < 0.05), with the Lr group being the lowest (0.16 g/L). Similar studies also showed that LAB degraded malic acid to lactic acid during the LAB fermentation, resulting in the increase of lactic acid content, i.e., malolactic fermentation (MLF) [19]. MLF could be utilized for decreasing the sourness of juices [30]. Shikimic acid is widely present in plants, but normally happens in low concentrations [19,32]. In this study, shikimic acid content in original orange juice was only 0.006 g/L, however, it was dramatically accumulated in the juice after the fermentation. Particularly in Lp, Bl, La and Lc samples, shikimic acid content was significantly increased to 0.051, 0.050, 0.043 and 0.037 g/L, respectively (*p* < 0.05). This phenomenon might be attributed to the shikimic acid pathway achieved by LAB using carbon sources during the fermentation [32]. Similar results have been reported on fermented bog bilberry juice [33] and apple juice [19].

#### 3.1.4. Variations in Vitamin C (Vc), Limonin and Flavones 

Vitamin C (Vc), as the most abundant water-soluble vitamin in orange juices, has a strong antioxidant and nutritional benefits for human health [34]. In the original orange juice (OJ), it was detected that the content of Vc was 59.27 mg/100 g. After fermentation, Vc contents in Lp, Lf, La and Bl samples were increased by 19.42%, 16.72%, 16.25% and 6.80% compared with the control, respectively, but those in Lr and Lc groups were reduced (Figure 3A). It might be because microorganisms could synthesize Vc during the fermentation process, leading to the increase of Vc content, while the decrease of Vc content could be related to the fact that it was easily degraded by chemical and enzymatic oxidation (such as ascorbate oxidase-induced microorganisms) [34]. 

Limonoids are crucial factors affecting the bitterness of orange juices, but they have various biological activities such as anti-tumor, antioxidative, antibacterial, etc. [35] Limonin is an important type of limonoids, as shown in Figure 3B, limonin content in all samples were significantly decreased at the end of fermentation (*p* < 0.05). This result showed that LAF could degrade limonin of orange juice. Additionally, it was found that the original navel orange contained a small amount of limonin, which was excellent to the processing and productions of orange juices.

Flavones, as one of the important quality factors of orange fruits, have various functions such as anti-inflammatory, antioxidant, as well as regulate color change and flavor development in fruits [36]. In this study, eight flavones were analyzed, in which three flavones (i.e., naringin, hesperidin, didymin) were detected without detectable neohesperidin, poncirin, sinensetin, rhoifolin, nobiletin (Figure 3C). Compared with the control, the flavone content of six fermented orange juices were decreased by 5.65–56.76%. Naringin is a representative bitter compound of the flavones in orange juice [37]. The content of naringin in each sample significantly decreased after fermentation (*p* < 0.05), which could help to reduce the bitterness of orange juices. These results were consistent with those from fermented orange wine [38] and apple juice [17], etc. 

### 3.2. Detection and Analysis on the Volatile Compounds of Fermented Orange Juice

Aromatic components or volatile compounds played a very essential role in orange juice quality [39]. A total of 43 volatile compounds were identified in the fermented orange juices and control. The quantified volatiles were grouped into eight chemical groups, including 16 sesquiterpenes, eight monoterpenes, seven alcohols, three esters, three ketones, three aldehydes, two acids and one piperazine (Table 2). Compared to the control (13 compounds), volatiles in Lp, Lc, Bl, La, Lf and Lr groups increased by 23, 21, 19, 19, 17, 15 compounds respectively, in total 30 new volatile compounds produced during fermentation, e.g., ethyl 3-hydroxyhexanoate, 2-carene, terpinolene, *α*-terpinene, *γ*-maaliene, etc.

The terpenes (monoterpenes and sesquiterpenes) were regarded as the predominant aromatic compounds in the volatile compounds of orange juice. As shown in Table 2, a total of 24 terpenes (eight monoterpenes and 16 sesquiterpenes) were detected in these samples, in which 18 types of new volatile compounds were increased in the fermented juices, probably due to new specific terpenes synthesized by some LAB [40], but the total contents of the terpenes reduced by 4.43–19.48% compared with the control. D-limonene was a representative volatile compound of the monoterpenes, which mainly provided citrus incense and mint incense of the fruits in *Citrus* [23]. After fermentation, the contents of D-limonene in six LAB fermentation strains were reduced by 2.20–53.85% compared with the control. In addition, monoterpenes such as 3-carene, (−)-4-terpineol and *α*-terpinene were accumulated during the fermentation, which imparted the lemon, pine, mint and anise incense of orange juice. Sesquiterpenes, as the second largest terpenes in orange juice, were increased by 33.62%, 45.78%, 48.70% and 96.71% in Lr, Bl, Lf and Lp groups, respectively. Valencene was the most abundant sesquiterpene in orange juice, which played an important role in contributing a strong green, citrus-like and oily aroma note [33]. In this study, the contents of valencene in Lf, Bl and Lr groups were relatively high as 108.91, 107.12 and 96.63 μg/mL, respectively. However, the high contents of terpenes could cause a spicy or bitter taste, which affected the feelings and preferences of consumers to the sweetness taste of juice [41]. Rosenfeld et al. also reported that the aroma of terpenoids contributed more to bitterness, which was mainly related to terpenes such as *α*-pinene, terpinolene and *β*-myrcene [41].

Besides the abundant terpenes, the alcohols also played an important role in the volatile compounds of orange juice. As presented in Table 2, the total alcohol contents in orange juices fermented with 6 LAB strains increased by 1.06–8.03 times compared to the control, which might be attributed to the degradation of glucose and the catabolism of amino acids [42]. For example, contents of (−)-4-terpineol and linalool in fermented orange juices obviously increased, which further enriched their sensory flavors such as rose and lavender incense, etc. Furthermore, a small amount of new volatile compounds, such as acids, esters, aldehydes, ketones were produced in the fermented orange juice (Table 2). The presence of aroma compounds, e.g., esters (ethyl 3-hydroxyhexanoate) and aldehydes (hexanal, pentanal, (Z)-citral), etc., enriched the fruity, grass flavors of orange juice (Table 3). Similar studies showed that low concentrations of aldehydes might cause great consumer acceptability [43].

A principal component analysis (PCA) was performed to visualize the relationships between fermented orange juices and control in terms of their volatile compounds (Figure 4A). As shown in Figure 4A, 58.1% of the total variability came from the first two principal components (PCs), in which PC1 accounted for 37.8% and PC2 accounted for 20.3%. Lp and La samples were located on the positive side of PC1, especially Lp sample was closely related to most of the aroma components such as sesquiterpenes and alcohols. Bl and Lf, which were clustered along the positive side of PC2, were closely related to aroma components such as ketones and small sesquiterpenes. Lc, Lr and control samples were clustered along the negative side of PC2 and closely related to terpenes and aldehydes, etc.

Following the PCA analysis, the heat map describing the volatile compounds showed the differences between 6 fermentations and control. As shown in Figure 4B, the heat map clearly exhibited their own peculiarities of volatiles, in which Lp group was mainly characterized by *β*-humulene, (+)-aromadendrene, *α*-terpinene, *α*-pinene, *α*-terpineol, (−)-4-terpineol, linalool, 1-octanol, citronellol, etc., Bl group by 3-hydroxyhexanoate and (Z)-citral, etc., Lf group by (+)-nootkatone, DL-alanine and (+)-longifolene, etc., Lc group by pentanal, *α*-guaiene and *β*-myrcene, etc., La group by 2-methyl piperazine, etc., Lr group by ethyl acetate and *γ*-maalinene, etc., and the control by *β*-myrcene, etc. Meanwhile, the results of the cluster showed that the orange juice fermented with Lp was pronouncedly separated from the other fermentations and control, highlighting the characteristics of the Lp strain.

Overall, LAB fermentation obviously influenced the contents and compositions of volatile compounds in orange juices, and improved their aroma flavors, whereas different LAB strains caused different effects. For example, Lp had a greater effect on the volatile compounds such as sesquiterpenes, alcohols, etc., while Lc and Lr strains had relatively little effect on the aroma composition such as monoterpenes (e.g., D-limonene, *β*-myrcene), at the same time, enriched esters, sesquiterpenes, aldehydes, and improved the sensory properties such as floral aroma and taste to a certain extent.

### 3.3. Detection and Analysis on Antioxidant Activity of the Fermented Orange Juice

Orange juice is rich in vitamin C, flavonoids, limonoids and carotenoids, etc., which has strong antioxidant activity and benefit to health. The DPPH RSA, ABTS RSA and FRAP were studied to assess antioxidant activities of fermented orange juices (Figure 5). After fermentation, the DPPH RSA in all fermentations (except of La) were increased to 63.58–80.25%, in which Lp sample had the highest DPPH RSA (80.25%). This result could be related to the *L. plantarum* fermentation significantly enhancing the utilization of antioxidant active ingredients such as Vc contents with proton donor properties [23]. In earlier studies, it was also found that LAF could indeed increase the availability of compounds with proton-donating properties, leading to an increase in DPPH RSA [47]. However, the DPPH RSA of La sample was only 46.86%, which decreased by 16.33% compared with the control (63.19%). The degradation and oxidation of antioxidant compounds might be the reason for the decrease of DPPH RSA, and different fermented strains posed different ability to use them [48,49]. Therefore, the increase of DPPH RSA in orange juice seemed to be closely related to the use of strain in the fermentation process [30,49].

As shown in Figure 5, ABTS RSA exhibited a significant increase in all fermentation samples (except of Lr) compared with the control (*p* < 0.05). Compared with other fermentation samples, ABTS RSA in Lp and La samples increased to 82.54% and 81.93% at the end of fermentation, which was 14.58% and 13.97% higher than that in control group (67.96%), respectively. The differences in ABTS RSA among different fermentations might be related to the influence degree of LAB on the antioxidant content of juice in the fermentation process [47]. However, it seemed that the ABTS RSA of juice samples as a whole has been improved after fermentation. Thus, LAF had a considerably positive effect on the ABTS radical scavenging activity in orange juice.

After fermentation, the FRAP of all fermentation samples were increased to 1.86–2.22 mmol Trolox/L, which were 15.53–37.89% higher than that in the original juice (1.61 mmol Trolox/L). And Bl sample had the best FRAP (peaking at 2.22 mmol Trolox/L), followed by Lf sample (2.12 mmol Trolox/L), Lp sample (2.08 mmol Trolox/L), and La sample (2.05 mmol Trolox/L). This phenomenon might be caused by some reducing agents produced during LAB fermentation, which reacted with free radicals and thus terminated the free radical chain reaction [15]. Therefore, we further confirmed that LAF could improve the reducing power of juices. In conclusion, the total antioxidant activity of orange juice fermented by *L. plantarum, L. fermentum*, *L. paracasei* extremely increased, especially after *L. plantarum* fermentation, which was higher than that of other strains. The result was similar to a previous study on the effects of *L. plantarum* fermentation on apple juice [19].

### 3.4. Analysis on the Sensory Properties of the Fermented Orange Juice

#### 3.4.1. Sensory Evaluation of the Fermented Orange Juice

In this study, the colors (orange color, light color), aroma flavors (green, orange aroma, fermented) and tastes (sourness, sweetness, bitterness) of fermented orange juice were evaluated with descriptive analysis (DA) method (Figure 6). Among the descriptors, higher average color scores of the fermented orange juices, including orange color (7.75), light color (8.13), acquired similar scores compared with control (8.0). Because the SSC/TA ratios in fermented orange juices significantly decreasing compared with the control (*p* < 0.05), it was found that the general satisfactions of fermented juices showed a downward trend, in which the sensory score (7.71) in the Lc sample was the greatest of all the fermented juices. The orange aroma of the fermented orange juices decreased to some extent compared with the control, due to the loss of D-limonene during the LAF, except with the Lc sample (Table 2). In addition, with the decrease of sugar content and the increase of organic acids as well as aroma types, clear variations were found in with regard to sourness, bitterness, green, and fermented flavor characteristics of fermented juices during the LAB fermentation. Of the six fermented juices, the Lc sample was the most popular with a pleasant fermented flavor (7.60), orange aroma (6.10) and sourness (8.20). The Bl sample was the second with consumers’ preferences on its light color (8.00) and fermented flavor (6.82). Furthermore, Lp, Lf and La samples exhibited their own flavor profiles, in which Lp sample had a strong sweet flavor (6.00), the Lf sample showed a strong green flavor (7.56), and the La sample was relatively bitter (3.80). It was further confirmed that LAF significantly influenced aroma and taste flavors of orange juices. 

#### 3.4.2. Analysis on the Aroma-Active Volatile Profiles Influencing Sensory Properties of Fermented Orange Juices

Aroma, as an important sensory characteristic of fruit products, directly affected the flavor quality and consumer acceptance of fermented juices [16]. Quantitative analysis of aroma compounds and calculation of odor activity value (OAV) could further confirm their contributions to fermented juices, and volatile compounds with OAV ≥ 1 were identified as aroma-active compounds [26]. The OAV of each aroma compound was calculated as the ratio of its concentration to its odor threshold, a total of 28 aroma-active compounds with OAV ≥ 1 was identified in this study (Table 3). As shown in Table 3, the result showed that the aroma-active compounds identified in Lf, La, Lr, Bl, Lc and Lp groups were 19, 19, 19, 20, 22, 25 species, respectively. In comparison, only 10 aroma-active compounds expressed OAV ≥ 1 in original orange juice (control). Among these aroma compounds, D-limonene of the control (OJ) was found to show the highest OAV (10,724), but its OAV in the fermented samples were decreased to 5537–10,489. This result further confirmed the conclusion that d-limonene was a representative aroma compound of the fruits in *Citrus* [23]. Twenty-two aroma-active compounds in Lc sample were detected, adding 12 new aroma-active compounds during fermentation compared to control, such as 3-carene (OAV, 152), pentanal (OAV, 11) and *β*-Caryophyllene (OAV, 3) with lemon, woody and almond flavors, etc. The other vital aroma-active compounds with OAV ≥ 1 in fermented orange juices and control were displayed in Table 3.

A study showed that odorants with high OAVs could significantly influence sensory properties of juices, and the higher the OAVs, the stronger the activity of aroma-active compounds [26]. As shown in Table 2 and Table 3, the contents of aroma-active compounds with high OAVs, e.g., D-limonene (OAV, 10,724, control), linalool (OAV, 4847, Lp) and ethyl butyrate (OAV, 2770, Lp), etc., significantly varied with the variations of fragrant odors such as orange aroma and flower incense before and after fermentation (*p* < 0.05), indicating that these aroma-active compounds had significant contribution to aroma flavors of the fermented orange juices. Thus, these aroma-active compounds with OAV ≥ 2000 such as D-limonene, linalool and ethyl butyrate were suggested as key aroma-active compounds influencing aroma flavors of the fermented orange juices. Similar studies also confirmed or suggested that limonene, linalool and ethyl butyrate were the major volatile compounds in orange juice and had important contributors to desirable flavor in orange products [38,39].

#### 3.4.3. Analysis on the Key Factors Significantly Influencing Sensory Properties of Fermented Orange Juice

In order to further analyze the contribution of the aroma-active compounds in the fermented samples to the aroma flavors, stepwise multiple regression analysis was performed to investigate the key aroma-active compounds significantly influencing the aroma flavors (green and fermented) of the fermented samples. The OAVs of 28 aroma-active compounds (OAV ≥ 1) were taken as the independent variables (*X*), while the aroma flavor of the orange juices was taken as the dependent variables (*Y*), respectively. The following regression equation model was obtained by regression analysis: *Y*_aroma flavor_ = 4.099 + 0.005 *X*_ethanol_ + 0.413 *X_β_*_-caryophyllene_.

As shown in Table 4, the generations of the models with no significant lack of fit implied their suitability. The *R*^2^ ≥ 90% in model 2 and variance analysis (sig. < 0.01) reflected that the model was well predictive. The results of the regression analysis showed that ethanol and *β*-caryophyllene were selected from 28 aroma-active compounds (factors) (*p* < 0.05), indicating the aroma flavors of the fermented orange juices were significantly influenced by ethanol and *β*-caryophyllene. Meanwhile, the results of Pearson correlation analysis showed that aroma flavors of the juices were positive significantly correlated with ethanol and *β*-caryophyllene (*p* < 0.05) (Appendix A), further confirming ethanol with sweet and apple incense, etc., and *β*-caryophyllene with woody, lilac incense, etc., had a great contribution to aroma flavors of fermented orange juices. Therefore, ethanol and *β*-caryophyllene were suggested as key aroma-active compounds significantly influencing aroma flavors of the fermented orange juices in this study. Moreover, As shown in Table 2 and Table 3, after the fermentation, ethanol contents in the fermented samples were significantly increased from 0 μg/mL (OAV, 0) to 0.56–3.19 μg/mL (OAV, 56–319), further enriching aroma flavors such as sweet and apple incense of the fermented orange juices. However, high ethanol content would produce strong pungent odor, affecting its aroma flavors of fermented orange juices. Thus, it was necessary to reasonably regulate ethanol content of fermented fruit juices during the LAF.

Similarly, in order to further analyze the contribution of the physicochemical characteristics to the taste quality of the fermented orange juices, including SSC, TA, SSC/TA ratio, pH, soluble sugars, organic acids, Vc, flavones, limonin and viable counts, stepwise multiple regression analysis was performed to investigate key physicochemical factors influencing the taste (sweetness and sourness) qualities of the fermented samples. The regression equation models were as follows:*Y*_sweetness_ = 7.268 − 0.284 *X*_organic acids_
*Y*_sourness_ = 11.183 − 0.432 *X*_SSC/TA ratio_

As shown in Table 5, the *R*^2^ (0.698, 0.973) and variance analysis (sig. < 0.05) reflected that the models were effective predictive. The results of the regression analysis showed that organic acids in model 1, SSC/TA ratio in model 2 were selected from 10 physicochemical factors (Table 5), indicating the sweetness flavor of fermented orange juice was significantly influenced by organic acids (*p* < 0.05), the sourness flavor significantly influenced by SSC/TA ratio (*p* < 0.01). Meanwhile, the result of Pearson correlation analysis showed that sweetness flavor of the juices was negative significantly correlated with organic acids (*p* < 0.05) (Appendix A), and the sourness was extremely significantly negative relation to SSC/TA ratio (*p* < 0.01) (Appendix A), further confirming organic acids, SSC/TA ratio had a significant effect on the sweetness and sourness of the fermented orange juices, respectively. Therefore, organic acids and SSC/TA ratio were suggested as key physicochemical factors significantly influencing taste flavors of the fermented orange juices in this study.

Generally, seven key factors significantly influencing sensory properties of the fermented orange juices were revealed in this study, including that ethanol and *β*-caryophyllene, significantly influencing aroma flavors (*p* < 0.05); SSC/TA ratio significantly influencing the sourness taste (*p* < 0.01); organic acids, significantly influencing the sweetness taste of the fermented orange juices (*p* < 0.05). Also, D-limonene, linalool and ethyl butyrate with high OAV (≥2000) were suggested as key aroma-active compounds significantly influencing aroma flavors of the fermented orange juices. These key factors played a decisive role in liking degree of consumers and qualities of fermented orange juices. 

## 4. Conclusions

In this study, we evaluated the effects of six LAB fermentations on the properties of orange juice. Our results showed that lactic acid fermentation could improve some bioactive compositions, aroma-active compounds, total antioxidant activities and sensory properties of orange juice. Seven key factors significantly influencing sensory properties of the juices were revealed from 28 aroma-active compounds and 10 physicochemical factors, including five key aroma compounds (ethanol, *β*-caryophyllene, D-limonene, linalool and ethyl butyrate) influencing aroma flavors, and two key physicochemical factors (SSC/TA ratio, organic acids) extremely influencing taste flavors, respectively. Of the six LAB fermentations, *L. paracasei* fermentation presented more desirable aroma flavors and overall acceptability, meanwhile enhancing antioxidant activities. This study could provide a reference for the deep processing of orange juices and the other fruit and vegetable juices.

## Figures and Tables

**Figure 1 foods-11-01920-f001:**
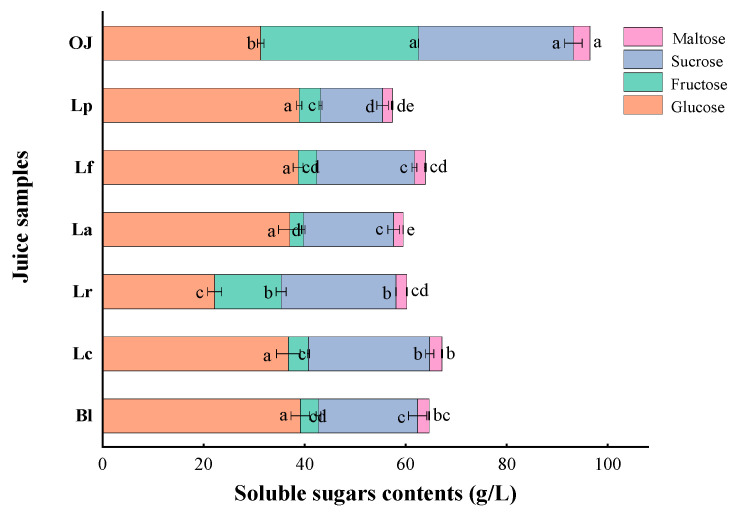
Soluble sugars contents of the orange juices fermented by *Lactiplantibacillus plantarum* (Lp), *L*. *fermentum* (Lf), *L*. *acidophilus* (La), *L*. *rhamnosus* (Lr), *L*. *paracasei* (Lc), *Bifidobacterium longum* (Bl) and control (*n* = 3). Control (OJ) was original orange juice. a–e Different letters in the same column indicate significant differences (*p* < 0.05).

**Figure 2 foods-11-01920-f002:**
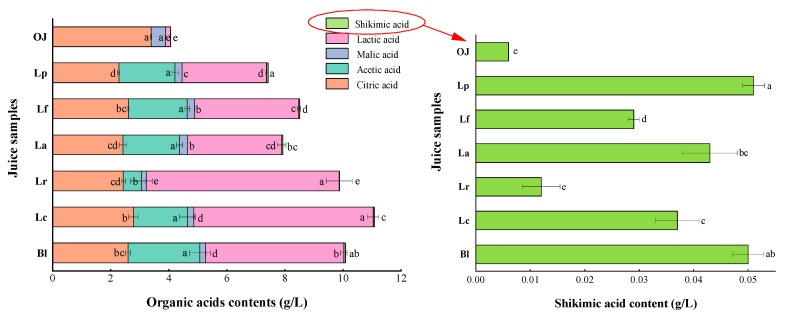
Organic acids contents of the orange juices fermented by *Lactiplantibacillus plantarum* (Lp), *L*. *fermentum* (Lf), *L*. *acidophilus* (La), *L*. *rhamnosus* (Lr), *L*. *paracasei* (Lc), *Bifidobacterium longum* (Bl) and control (*n* = 3). Control (OJ) was original orange juice. a–e Different letters in the same column indicate significant differences (*p* < 0.05).

**Figure 3 foods-11-01920-f003:**
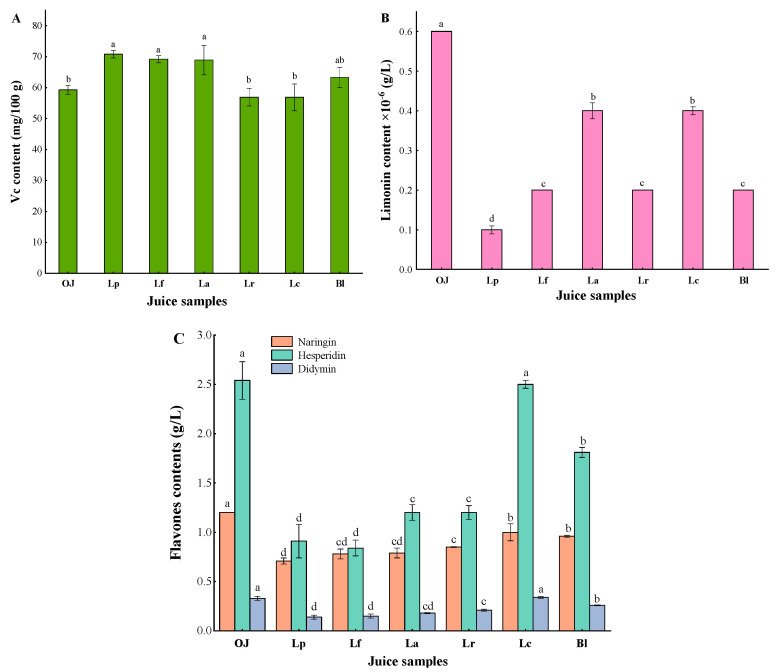
Vc (**A**), limonin (**B**) and flavones contents (**C**) of the orange juices fermented by *Lactiplantibacillus plantarum* (Lp), *L*. *fermentum* (Lf), *L*. *acidophilus* (La), *L*. *rhamnosus* (Lr), *L*. *paracasei* (Lc), *Bifidobacterium longum* (Bl) and control (*n* = 3). Control (OJ) was original orange juice. a–d Different letters in the same column indicate significant differences (*p* < 0.05).

**Figure 4 foods-11-01920-f004:**
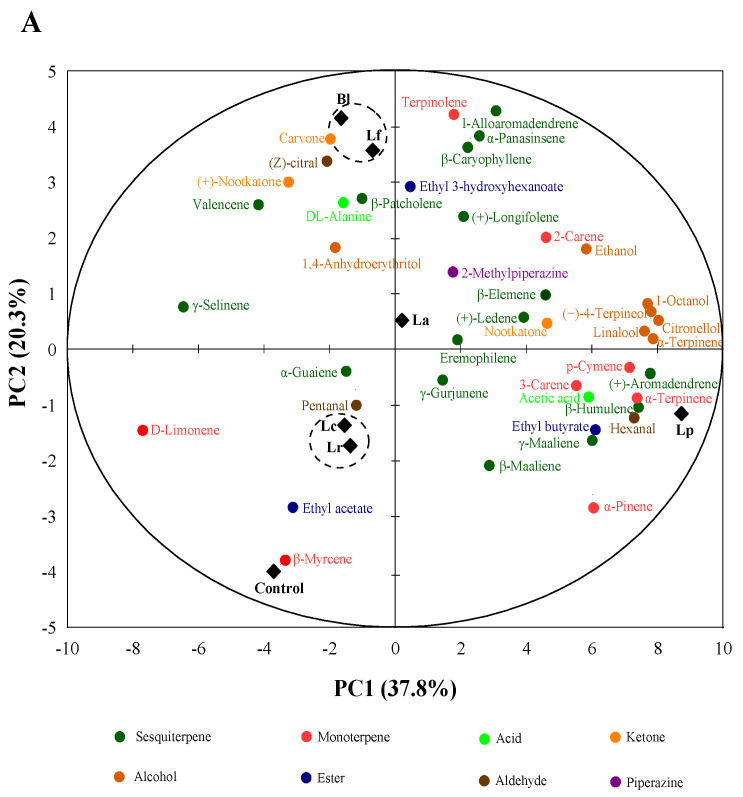
Principal component analysis: (**A**); and heat map (**B**) to evaluate the orange juices fermented by *Lactiplantibacillus plantarum* (Lp), *L*. *fermentum* (Lf), *L*. *acidophilus* (La), *L*. *rhamnosus* (Lr), *L*. *paracasei* (Lc), *Bifidobacterium longum* (Bl) and control (*n* = 3). Control (OJ) was original orange juice.

**Figure 5 foods-11-01920-f005:**
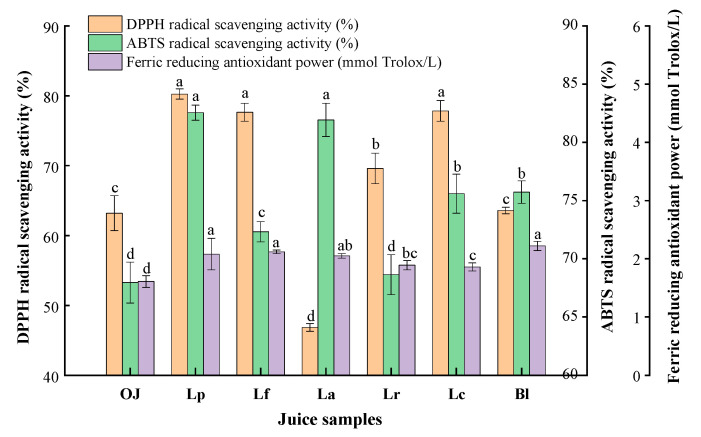
Antioxidant activities of the orange juices fermented by *Lactiplantibacillus plantarum* (Lp), *L*. *fermentum* (Lf), *L*. *acidophilus* (La), *L*. *rhamnosus* (Lr), *L*. *paracasei* (Lc), *Bifidobacterium longum* (Bl) and control (*n* = 3). Control (OJ) was original orange juice. a–d Different letters in the same column indicate significant differences (*p* < 0.05).

**Figure 6 foods-11-01920-f006:**
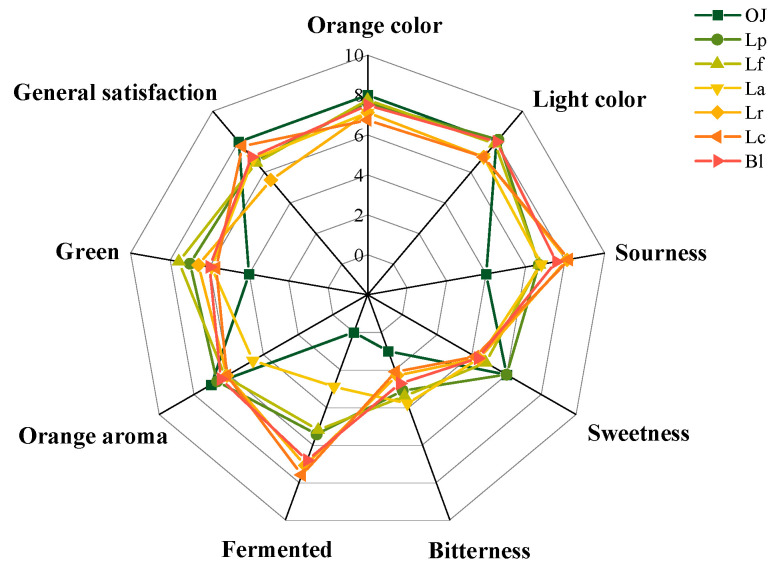
Sensory radar for the mean values of the sensory attributes of the orange juices fermented by *Lactiplantibacillus plantarum* (Lp), *L. fermentum* (Lf), *L. acidophilus* (La), *L. rhamnosus* (Lr), *L. paracasei* (Lc), *Bifidobacterium longum* (Bl) and control (*n* = 3). Control (OJ) was original orange juice.

**Table 1 foods-11-01920-t001:** Physicochemical characteristics and viable counts in the fermented orange juices and control (*n* = 3).

Index	OJ	Lp	Lf	La	Lr	Lc	Bl
Viable counts (log CFU/mL)	-	7.48 ± 0.24 b	7.58 ± 0.59 ab	7.42 ± 0.01 b	7.71 ± 0.26 ab	8.07 ± 0.04 a	7.84 ± 0.32 ab
pH	6.60 ± 0.39 a	4.82 ± 0.02 b	4.17 ± 0.02 cd	4.23 ± 0.12 cd	3.96 ± 0.12 d	4.33 ± 0.02 c	4.48 ± 0.14 c
SSC (°Brix)	11.24 ± 0.25 a	8.20 ± 0.30 c	8.85 ± 0.07 b	8.17 ± 0.31 c	8.05 ± 0.07 c	8.20 ± 0.30 c	8.80 ± 0.26 b
TA (%)	0.69 ± 0.01 c	0.76 ± 0.03 bc	0.79 ± 0.03 bc	0.86 ± 0.03 b	1.09 ± 0.07 a	1.23 ± 0.01 a	1.14 ± 0.00 a
SSC/TA ratio	16.30 ± 0.51 a	10.75 ± 0.34 b	11.12 ± 0.00 b	9.64 ± 1.31 b	7.69 ± 0.12 c	6.70 ± 0.86 c	7.79 ± 0.83 c

The pH of OJ was adjusted to 6.60 ± 0.39. Orange juices fermented by *Lactiplantibacillus plantarum* (Lp), *L. fermentum* (Lf), *L. acidophilus* (La), *L. rhamnosus* (Lr), *L. paracasei* (Lc), *Bifidobacterium longum* (Bl) and control (*n* = 3). Control (OJ) was original orange juice. SSC: soluble solid content. TA: total acid. a–d Different letters in the same row indicate significant differences (*p* < 0.05).

**Table 2 foods-11-01920-t002:** Quantitative concentrations (μg/mL) of volatile compounds in fermented orange juices and control (*n* = 3).

No.	Compounds	Control	Fermented Orange Juices					
			Lp	Lf	La	Lr	Lc	Bl
	Esters							
1	Ethyl acetate	0.32 ± 0.15 a	ND	ND	ND	0.51 ± 0.00 a	ND	ND
2	Ethyl butyrate	1.19 ± 0.51 b	2.27 ± 0.18 a	0.77 ± 0.13 bc	1.17 ± 0.43 b	0.19 ± 0.04 d	0.52 ± 0.12 cd	0.35 ± 0.02 cd
3	Ethyl 3-hydroxyhexanoate	ND	0.86 ± 0.27 b	0.43 ± 0.19 bc	0.13 ± 0.05 c	0.94 ± 0.31 b	0.12 ± 0.00 c	2.99 ± 0.31 a
	Monoterpenes							
4	*α*-Pinene	1.30 ± 0.37 b	2.31 ± 0.28 a	0.40 ± 0.07 cd	0.35 ± 0.05 cd	0.59 ± 0.08 cd	0.67 ± 0.13 c	0.28 ± 0.13 d
5	*β*-Myrcene	8.73 ± 0.04 a	5.01 ± 1.09 c	4.69 ± 1.39 c	5.07 ± 1.49 c	5.97 ± 1.47 bc	8.25 ± 0.70 ab	4.16 ± 1.23 c
6	D-Limonene	364.62 ± 6.87 a	188.27 ± 15.61 d	281.61 ± 30.12 c	308.71 ± 9.52 bc	336.9 ± 24.43 ab	356.61 ± 28.95 a	306.88 ± 5.90 bc
7	2-Carene	ND	0.43 ± 0.05 a	0.09 ± 0.01 c	0.43 ± 0.07 a	0.21 ± 0.07 b	0.17 ± 0.06 bc	0.47 ± 0.05 a
8	3-Carene	ND	0.95 ± 0.14 a	0.59 ± 0.08 c	0.62 ± 0.06 bc	0.69 ± 0.05 bc	0.76 ± 0.00 b	ND
9	Terpinolene	ND	0.27 ± 0.04 ab	0.55 ± 0.31 a	0.19 ± 0.09 ab	0.13 ± 0.08 b	0.23 ± 0.14 ab	0.37 ± 0.29 ab
10	*α*-Terpinene	ND	0.41 ± 0.26 a	ND	0.05 ± 0.00 b	0.09 ± 0.07 b	0.12 ± 0.05 b	0.07 ± 0.01 b
11	p-Cymene	ND	0.94 ± 0.64 a	ND	0.73 ± 0.12 a	ND	ND	ND
	Sesquiterpenes							
12	Valencene	82.43 ± 0.34 b	55.03 ± 2.26 c	108.91 ± 4.41 a	62.57 ± 2.06 c	96.63 ± 4.68 a	60.27 ± 1.97 c	107.12 ± 9.41 a
13	Eremophilene	1.21 ± 0.02 c	1.84 ± 0.00 a	1.77 ± 0.12 ab	ND	1.70 ± 0.11 ab	1.49 ± 0.33 abc	1.47 ± 0.09 bc
14	*γ*-Maaliene	ND	7.46 ± 0.79 a	ND	ND	5.90 ± 1.30 a	ND	ND
15	(+)-Ledene	ND	1.98 ± 0.00 a	2.31 ± 0.21 a	ND	2.06 ± 0.19 a	ND	ND
16	*β*-Elemene	ND	0.49 ± 0.04 ab	0.38 ± 0.03 bc	0.48 ± 0.04 ab	0.50 ± 0.06 a	0.44 ± 0.10 ab	0.27 ± 0.04 c
17	*β*-Caryophyllene	ND	0.55 ± 0.14 ab	0.64 ± 0.09 ab	0.47 ± 0.05 b	0.56 ± 0.17 ab	0.54 ± 0.10 ab	0.76 ± 0.08 a
18	*β*-Maaliene	ND	0.97 ± 0.00 b	ND	ND	1.67 ± 0.01 a	0.63 ± 0.27 b	ND
19	*γ*-Gurjunene	ND	0.81 ± 0.13 b	0.84 ± 0.01 b	1.48 ± 0.46 a	1.53 ± 0.00 a	1.19 ± 0.26 ab	0.07 ± 0.01 c
20	*γ*-Selinene	5.08 ± 0.01 cd	0.47 ± 0.00 e	7.77 ± 0.77 a	4.59 ± 0.64 cd	6.83 ± 0.83 ab	6.03 ± 1.67 bc	4.05 ± 0.06 d
21	(+)-Longifolene	ND	0.51 ± 0.00 b	1.41 ± 0.51 a	0.04 ± 0.00 c	0.31 ± 0.00 bc	ND	0.15 ± 0.01 c
22	*α*-Panasinsanene	ND	4.76 ± 0.73 ab	6.24 ± 1.21 ab	4.66 ± 0.71 ab	ND	4.52 ± 1.56 b	6.59 ± 0.56 a
23	(+)-Aromadendrene	ND	1.36 ± 0.19 a	0.10 ± 0.00 b	ND	ND	ND	0.04 ± 0.00 c
24	Alloaromadendrene	ND	1.29 ± 0.00 ab	1.56 ± 0.28 ab	1.03 ± 0.15 ab	0.86 ± 0.84 b	0.83 ± 0.10 b	1.76 ± 0.24 a
25	*β*-Patchoulene	ND	ND	ND	5.90 ± 0.21 b	ND	ND	7.06 ± 0.56 a
26	*β*-Humulene	ND	0.97 ± 0.10	ND	ND	ND	ND	ND
27	*α*-Guaiene	ND	ND	ND	0.04 ± 0.00 b	ND	0.07 ± 0.00 a	ND
	Acids							
28	Acetic acid	ND	4.47 ± 0.60 a	ND	4.40 ± 0.15 a	ND	2.63 ± 0.03 b	ND
29	DL-Alanine	ND	ND	0.41 ± 0.01 a	ND	ND	0.23 ± 0.02 b	0.11 ± 0.01 c
	Alcohols							
30	Linalool	0.90 ± 0.01 e	7.27 ± 0.80 a	3.06 ± 0.59 c	4.24 ± 0.66 b	2.41 ± 0.19 cd	1.32 ± 0.31 e	1.80 ± 0.08 de
31	(−)-4-Terpineol	1.55 ± 0.05 e	10.19 ± 0.92 a	4.78 ± 0.15 b	5.38 ± 0.41 b	3.41 ± 0.39 c	2.29 ± 0.15 de	2.99 ± 0.30 cd
32	*α*-Terpineol	0.15 ± 0.08 b	2.14 ± 1.16 a	0.62 ± 0.46 b	0.73 ± 0.13 b	0.43 ± 0.36 b	0.15 ± 0.04 b	0.50 ± 0.12 b
33	Citronellol	ND	0.51 ± 0.00 a	0.14 ± 0.06 b	0.05 ± 0.01 c	0.11 ± 0.02 bc	0.05 ± 0.00 c	0.16 ± 0.06 b
34	Ethanol	ND	2.90 ± 0.21 a	3.19 ± 0.74 a	1.23 ± 0.05 b	0.59 ± 0.02 c	0.59 ± 0.04 c	0.56 ± 0.03 c
35	1-Octanol	ND	0.47 ± 0.00 a	0.16 ± 0.17 b	0.09 ± 0.05 b	ND	0.14 ± 0.00 b	0.10 ± 0.03 b
36	1,4-Anhydroerythritol	ND	ND	ND	0.78 ± 0.83 a	ND	0.81 ± 0.33 a	0.94 ± 0.22 a
	Ketones							
37	Carvone	0.64 ± 0.05 b	0.21 ± 0.14 bc	2.21 ± 0.15 a	2.34 ± 0.29 a	0.52 ± 0.21 bc	0.06 ± 0.02 c	2.25 ± 0.30 a
38	Nootkatone	ND	0.42 ± 0.04 a	ND	0.11 ± 0.03 c	ND	0.37 ± 0.00 a	0.32 ± 0.01 b
39	(+)-Nootkatone	0.18 ± 0.02 c	0.01 ± 0.00 e	0.58 ± 0.10 a	0.11 ± 0.04 d	0.31 ± 0.02 b	0.03 ± 0.03 de	0.36 ± 0.00 b
	Aldehydes							
40	Hexanal	ND	0.33 ± 0.23	ND	ND	ND	ND	ND
41	Pentanal	ND	ND	ND	ND	ND	0.13 ± 0.02	ND
42	(Z)-citral	ND	ND	0.04 ± 0.00 c	ND	ND	0.05 ± 0.00 b	0.14 ± 0.02 a
	Piperazines							
43	2-Methylpiperazine	ND	0.13 ± 0.04 bc	0.20 ± 0.07 b	0.57 ± 0.05 a	ND	0.02 ± 0.01 d	0.06 ± 0.01 cd

Orange juices fermented by *Lactiplantibacillus plantarum* (Lp), *L. fermentum* (Lf), *L. acidophilus* (La), *L. rhamnosus* (Lr), *L. paracasei* (Lc), *Bifidobacterium longum* (Bl) and control (*n* = 3). Control (OJ) was original orange juice. ND: not detected. a–e Different letters in the same row indicate significant differences (*p* < 0.05).

**Table 3 foods-11-01920-t003:** OAVs of aroma-active compounds with OAV ≥1 in fermented orange juices and control (*n* = 3).

No.	Compound	OT ^a^	OAV ^b^							Odor Quality ^c^
			OJ	Lp	Lf	La	Lr	Lc	Bl	
1	Ethyl acetate	0.1	ND	ND	ND	ND	5.1	ND	ND	Pineapple
2	Ethyl butyrate	0.001	1190	2770	770	1170	190	520	350	Apple
3	Ethyl3-hydroxyhexanoate	0.01	ND	86	43	13	94	12	299	Fruity
4	*α*-Pinene	0.033	39	70	12	11	18	20	8	Pine, turpentine
5	*β*-Myrcene	0.05	174	100	94	101	119	165	83	Spice, must, balsamic
6	D-Limonene	0.034	10,724	5537	8283	9080	9909	10,489	9026	Citrus, mint
7	2-Carene	0.006	ND	72	15	72	35	28	78	Turpentine
8	3-Carene	0.005	ND	190	118	124	138	152	ND	Lemon, resin, citrus
9	Terpinolene	0.2	ND	1	3	<1	<1	1	2	Pine, plastic
10	α-Terpinene	0.056	ND	7	ND	<1	2	2	1	Lemon, mint, oil
11	p-Cymene	0.0133	ND	71	ND	55	ND	ND	ND	Citrus, gasoline
12	Valencene	10.5	8	5	10	6	9	6	10	Green, oil, orange-like
13	*β*-Caryophyllene	0.16	ND	3	4	3	4	3	5	Sweet, woody, lilac
14	*β*-Humulene	0.16	ND	6	ND	ND	ND	ND	ND	Woody, balsamic
15	*γ*-Selinene	0.036	141	13	216	128	190	168	113	Woody, herb
16	Acetic acid	0.013	ND	344	ND	338	ND	202	ND	Sour, pungent
17	Linalool	0.0015	600	4847	2040	2827	1607	880	1200	Flower, lavender
18	(−)-4-Terpineol	0.4	4	25	12	13	9	6	7	Rose
19	*α*-Terpineol	0.3	<1	7	2	2	1	<1	2	Pine, terpene, lilac
20	Citronellol	0.01	ND	51	14	5	11	5	16	Rose
21	Ethanol	0.01	ND	290	319	123	59	59	56	Sweet, apple, pungent
22	1-Octanol	0.11	ND	4	1	<1	ND	1	<1	Moss, nut, mushroom
23	Carvone	0.086	7	2	26	27	6	<1	26	Mint, basil, fennel
24	Nootkatone	0.18	ND	2	ND	<1	ND	2	2	Grapefruit
25	(+)-Nootka tone	0.001	180	10	580	110	310	30	360	Orange-like
26	Hexanal	0.005	ND	66	ND	ND	ND	ND	ND	Grass, tallow, fat
27	Pentanal	0.012	ND	ND	ND	ND	ND	11	ND	Almond, malt
28	(Z)-citral	0.05	ND	ND	<1	ND	ND	1	3	Lemon

^a^ OT: odor threshold (μg/mL). ^b^ OAV: odor activity value, the ratio of the concentration of a volatile aroma component to its odor threshold value. All the odor thresholds were obtained from [44,45,46]. ^c^ All the odor descriptions were cited from [40,44] and www.flavornet. org. Orange juices fermented by *Lactiplantibacillus plantarum* (Lp), *L. fermentum* (Lf), *L. acidophilus* (La), *L. rhamnosus* (Lr), *L. paracasei* (Lc), *Bifidobacterium longum* (Bl) and control (*n* = 3). Control (OJ) was original orange juice. ND: not detected.

**Table 4 foods-11-01920-t004:** Stepwise multiple regression analysis on the key factors of volatile compounds with OAV ≥ 1 significantly influencing aroma flavors of the fermented orange juices and original orange juice (control).

Model	Factors	Regression Coefficients	*t*-Value	*p*-Value	*R* ^2^	*F*-Value	Sig.
1	ethanol	5.136 (constant)	12.560	0.000	0.667	10.006	0.025
		0.007	3.163	0.025			
2		4.099 (constant)	10.570	0.000	0.913	20.875	0.008
	ethanol	0.005	3.679	0.021			
	*β*-caryophyllene	0.413	3.353	0.028			

**Table 5 foods-11-01920-t005:** Stepwise multiple regression analysis on the key factors of the physicochemical characteristics and viable counts significantly influencing taste qualities of the fermented orange juices and original orange juice (control).

Model	Taste Quality (*Y*)	Factors (*X*)	Regression Coefficients	*t*-Value	*p*-Value	*R* ^2^	*F*-Value	Sig.
1	sweetness	organic acid	7.268 (constant)	10.028	0.000	0.698	11.560	0.019
			−0.284	−3.400	0.019			
2	sourness	SSC/TA ratio	11.183 (constant)	33.319	0.000	0.973	180.776	0.000
			−0.432	−13.445	0.000			

SSC/TA ratio: soluble solid content/ total acid content ratio.

## Data Availability

The data are avaliable from the corresponding author.

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
