# Peer review of "Effect of Six Lactic Acid Bacteria Strains on Physicochemical Characteristics, Antioxidant Activities and Sensory Properties of Fermented Orange Juices"

_foods, 2022, doi:10.3390/foods11131920_

Round 1

Reviewer 1 Report

Comments and Suggestions for Authors

Quan and others investigate the effects of the 6 LAB strain on the properties of fermented orange juice. Using various important indicators for evaluating the quality of the beverage, the authors suggested that the use of the LAB strains can greatly improve the quality of the fermented orange juice. The approach to evaluate the properties seems to be reasonable, and the conclusion is also appropriate based on the sufficient results.

Though the manuscript was well described, some parts should be improved.

1) The discussion of shikimic acid should be described more as those for other organic acids were written properly.

2) The discussion of antioxidant activities should be described in more detail. The role of LAB strains in antioxidant activity was just written by citing other previous reports. In addition, the reasons why DPPH RSA of La sample was lower than all other samples were not discussed.

3) It is necessary to add the information on the safety assessment of the tested LAB strains (seems appropriate to describe it in the Materials and Methods section) although the species of the LAB are well-known as GRAS. If the assessment is not tested, please describe the necessity to test the safety of the tested LAB strains.

Other minor comments

1) Please revise the genus names of LAB as the taxonomy of the Lactobacillaceae has been updated since 2020. Please refer to the following paper.

Zheng, J., Wittouck, S., Salvetti, E., Franz, C. M., Harris, H., Mattarelli, P., ... & Lebeer, S. (2020). A taxonomic note on the genus Lactobacillus: Description of 23 novel genera, emended description of the genus Lactobacillus Beijerinck 1901, and union of Lactobacillaceae and Leuconostocaceae.

2) Line 289: Please add space after “decreased to”.

3) Line 349: Please inactivate the Bold function for terpinolene.

4) Lines 702-703: Please revised the reference format.

Author Response

Please refer to the attachment, thank you.

Reviewer 2 Report

The manuscript (Effect of 6 Lactic Acid Bacteria Strains on Physicochemical 2 Characteristics, Antioxidant Activities and Sensory Properties 3 of Fermented Orange Juices) has good idea but it needs to  many  corrects and modifications .

1-From the rules of the language, numbers from 1-9 should be written in a written form, not a number like 6 write six.

2- The title of the manuscript contains the number 6, it should be corrected to six.

3-The names of lactic acid bacteria should be written in the new nomenclature throughout the manuscript. Lactobacillus plantarum, for example, is written as Lactiplantibacillus plantarum.

4-Page 2 line 60-62 , this section needs to add references, I suggest new reference in here (Al-Sahlany, S. T., & Niamah, A. K. (2022). Bacterial viability, antioxidant stability, antimutagenicity and sensory properties of onion types fermentation by using probiotic starter during storage. Nutrition & Food Science.)

5-Page 3 line 116, What is the number of bacteria present in this size of the bacterial inoculum?

6- Page 3 , line 122, Viable Cell Counts method need to add new reference, I suggest (Peng, W., Meng, D., Yue, T., Wang, Z., & Gao, Z. (2021). Effect of the apple cultivar on cloudy apple juice fermented by a mixture of Lactobacillus acidophilus, Lactobacillus plantarum, and Lactobacillus fermentum. Food Chemistry, 340, 127922.

7- Page 3 line 124 , what do you mean of AGAR pour method ?

8-Page 3 line 129, Total Acids (TA) need add the reference.

9-Table 1 page 6, It is scientifically known that an increase in acidity leads to a decrease in pH. The results of the total acidity are inversely proportional to the results of the pH, but we notice that at pH 4.17 the value of the total acidity was 0.79 in the (Lf) sample, while in the BI sample the acidity was 1.14 when the pH was 4.48. How do you explain this discrepancy in the results???

10-Figure 3 and 5 need to add a title in the x-axisز

11-The conclusions in the manuscript contain many results, the conclusions should be rewritten.‏

Author Response

(The authors gave the same response as above.)

Round 2

Reviewer 1 Report

Though there are several minor errors, the production Editor or authors can revise during the proofreading process.

Reviewer 2 Report

Dear Editor (s),

The authors made all the necessary changes to improve the manuscript